# Full-cycle device-scale simulations of memory materials with a tailored atomic-cluster-expansion potential

Yuxing Zhou [1], Daniel F. Thomas du Toit[1], Stephen R. Elliott[2], Wei Zhang [3] ✉ & Volker L. Deringer [1] ✉

Computer simulations have long been key to understanding and designing phase-change materials (PCMs) for memory technologies. Machine learning is now increasingly being used to accelerate the modelling of PCMs, and yet it remains challenging to simultaneously reach the length and time scales required to simulate the operation of real-world PCM devices. Here, we show how ultra-fast machine-learned interatomic potentials, based on the atomic cluster expansion (ACE) framework, enable simulations of PCMs reflecting applications in devices with excellent scalability on high-performance computing platforms. We report full-cycle simulations—including the time-consuming crystallisation process (from digital "zeroes" to "ones")—thus representing the entire programming cycle for cross-point memory devices. We also showcase a simulation of full-cycle operations, relevant to neuromorphic computing, in a mushroom-type device geometry.

Phase-change materials (PCMs) from the Ge–Sb–Te system have been widely used in emerging electronic devices, including non-volatile memory and neuromorphic in-memory computing technologies[1–6]. Driven by Joule heating resulting from the application of electric pulses, the SET (crystallisation) and RESET (amorphisation) operations are associated with fast and reversible transitions between the amorphous (low-conductance) and crystalline (high-conductance) states of PCMs. A large property contrast between these states encodes "zeroes" and "ones" in the atomic structure, respectively, for binary memory[7]. Furthermore, finely tuning the conductance of PCM cells between all-amorphous and all-crystalline states enables multi-level programming for neuromorphic in-memory computing[8].

The switching processes in PCMs can be completed within nanoseconds—that is, within time scales accessible for molecular-dynamics (MD) computer simulations—and this has long made PCMs a prime application area in the field of materials modelling. Density-functional theory (DFT)-driven ab initio molecular-dynamics (AIMD) simulations have played a key role in understanding structural features[9–11], property contrast[12–15], and crystallisation kinetics[16–18] of

PCMs based on representative small-scale models (typically containing on the order of 1000 atoms or fewer)[18]. Building on the long-standing successes of DFT and AIMD, machine-learning (ML)-based interatomic potentials have recently emerged which accelerate first-principles atomistic modelling by many orders of magnitude[19–21], and which can therefore provide new insights at much-extended time and length scales.

More than a decade ago already, Sosso et al. reported the first ML potential for modelling PCMs, at that time for the binary compound GeTe[22], based on the Behler–Parrinello neural-network framework[23]. Since then, ML-driven MD simulations have become gradually more established: for example, they revealed details of the temperature-dependent crystallisation in GeTe[24] and $Ge_2Sb_2Te_5$ alloys[25]. In time, more ML potentials have begun to be developed for different PCMs[25–29]. We recently reported a chemically transferable and defect-tolerant ML potential for Ge–Sb–Te (GST) materials along the GeTe–$Sb_2Te_3$ tie-line, fitted using the Gaussian approximation potential (GAP) framework[30] and based on a comprehensive structural dataset and iterative training[31]. A neuro-evolution potential was

[1]Inorganic Chemistry Laboratory, Department of Chemistry, University of Oxford, Oxford, UK. [2]Physical and Theoretical Chemistry Laboratory, Department of Chemistry, University of Oxford, Oxford, UK. [3]Center for Alloy Innovation and Design (CAID), State Key Laboratory for Mechanical Behavior of Materials, Xi'an Jiaotong University, Xi'an, China. ✉e-mail: wzhang0@mail.xjtu.edu.cn; volker.deringer@chem.ox.ac.uk

recently developed for large-scale crystallisation simulations of $Sb_2Te$, SbTe, and $Sb_2Te_3$, revealing distinct behaviours driven by nucleation and growth[32]. More generally, graph-based ML methods constitute the current state-of-the-art architectures in terms of accuracy and chemical transferability[33,34], and they have begun to attract attention in the PCM community for general-purpose simulation tasks[35].

In 2025, computational modelling is now often able to describe "the real thing" thanks to ML-driven potentials[36], and yet these models still face a significant obstacle when it comes to describing PCMs in a fully realistic way—e.g., because of the length scales and structural complexity associated with applications in this domain. Complex simulation protocols are therefore required to model PCM devices, such as non-isothermal heating, which we have demonstrated for both cross-point and mushroom-type cells[31]. Using the GST-GAP-22 potential at the time, we simulated a 50-picosecond RESET operation ("1 → 0"), showing non-isothermal melting and rapid cooling in a 532,980-atom model, representing a cell volume of $20 \times 20 \times 40$ nm³ in cross-point memory devices[31]. However, the subsequent SET ("0 → 1") typically requires tens of nanoseconds, rather than tens of picoseconds, to complete in devices. Performing a crystallisation run over 10 ns for the same structural model, with GST-GAP-22, would have consumed more than 150 million CPU core hours by our estimate. This type of excessive cost (in terms of time, financial cost, and carbon emissions) would clearly make the use of GST-GAP-22 unfeasible for full-cycle modelling of GST devices.

Herein, we show how one can simultaneously reach both the length and time scales in simulations of switching operations in real-world GST devices, leveraging the atomic-cluster-expansion (ACE) ML framework[37]. The substantial speed-up by moving from the GAP to the ACE framework[38] enables atomistic simulations reflecting device applications on widely available CPU-based computing systems. We have thus outlined an "off-the-shelf"-usable ML approach for the community to study the switching mechanisms of GST-based devices. Beyond PCMs, our work explores the current frontiers of ultra-large-scale all-atom simulations for materials science and engineering.

## Results

### Fast and CPU-efficient simulations with an optimised ACE potential

We used the ACE framework to develop a computationally efficient ML model for GST. In ACE, the local environment of a given atom is encoded using a many-body expansion (Fig. 1a). The atomic

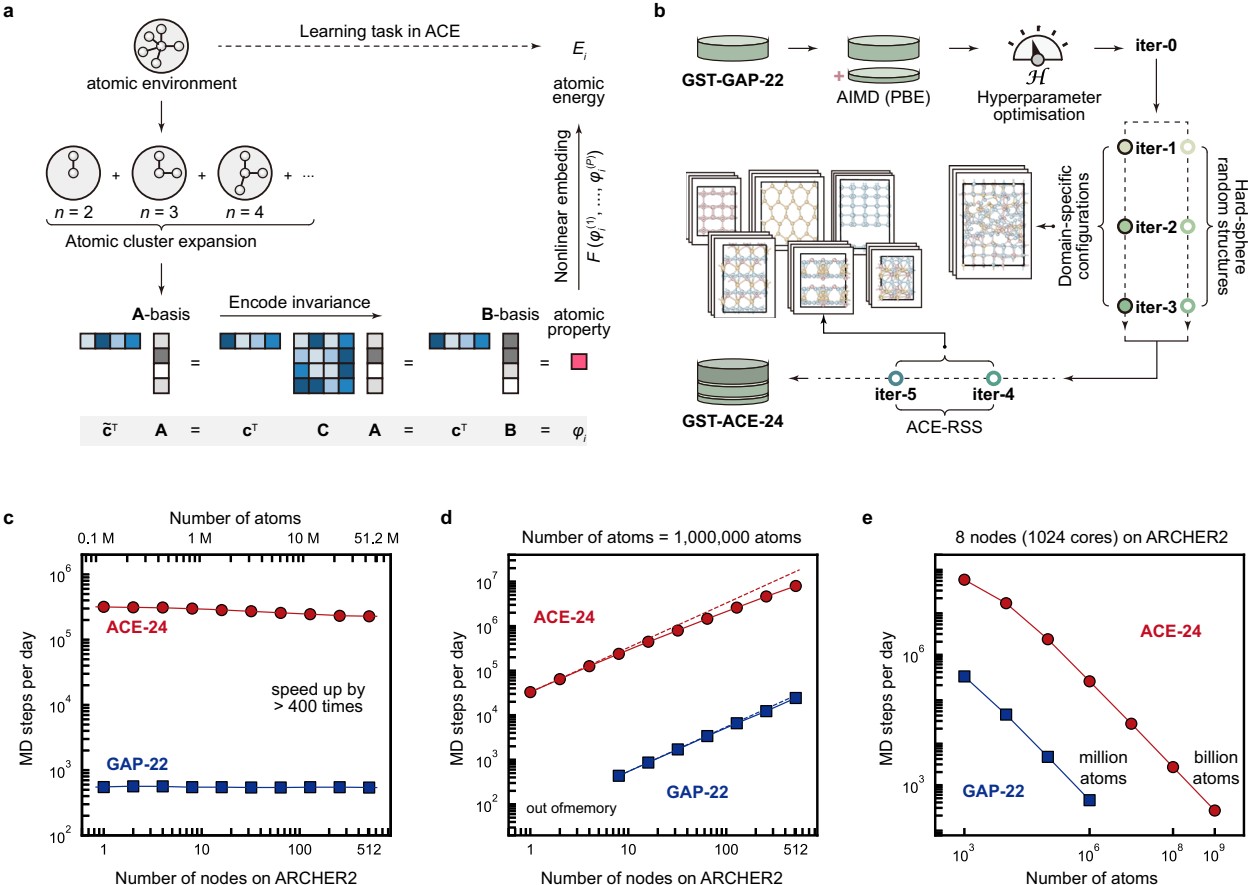

**Fig. 1 | An optimised atomic-cluster-expansion (ACE) machine-learning (ML) potential for Ge−Sb−Te (GST) phase-change materials. a** Schematic overview of the learning task in ACE, showing the matrix equations that define the fitting of ACE models. The atomic property, $\varphi$, of a given atomic environment is expressed using a linear combination (via the coefficient vector $\tilde{c}$) of "A-basis" functions, **A**, or using a linear combination (via the coefficient vector $c$) of invariant "B-basis" functions, **B**, by coupling via the generalised Clebsch−Gordan coefficients, **C**, i.e., $\tilde{c} = c \cdot C$. The energy of the atom, $E$, is predicted as a function of various atomic properties, using a linear or nonlinear embedding, i.e., $E = F(\varphi_i^{(1)}, \ldots, \varphi_i^{(P)})$. **b** The fitting protocol used for a new ACE potential of GST. Multistep iterations were carried out to expand the reference dataset. Insets show typical configurations for domain-specific iterations and ACE-driven random-structure searching (ACE-RSS) structures. Ge, Sb, and Te atoms are rendered as light red, light yellow, and light blue, respectively. Details are given in Supplementary Note 1. Different tests of computational efficiency on the ARCHER2 high-performance computing system were carried out for the GST-ACE-24 (this work) and GST-GAP-22[31] potentials, including: **c** weak scaling (at 100,000 atoms per node); **d** strong scaling up to 512 nodes−the maximum accessible for a single job on ARCHER2−for a 1-million-atom structural model; and (**e**) molecular-dynamics (MD) steps per day as a function of system size running on 8 computing nodes (1024 CPU cores). The GAP-MD tests in (**d**) failed on configurations with 4 or fewer nodes due to insufficient memory. The dashed lines in (**d**) indicate ideal strong scaling behaviour.

environment is expressed in terms of radial functions and spherical harmonics, translated into a linear combination of so-called "A-basis" functions, and subsequently into invariant "B-basis" functions by coupling via the generalised Clebsch–Gordan coefficients. A linear combination of B-basis functions is called the "property" of a given atomic environment in the context of ACE. The energy of the atom is predicted as a function of atomic properties, using a linear (depending on just one single property) or nonlinear embedding. The complexity of ACE models is therefore controlled by the numbers of basis and embedding functions; more details of the framework can be found in refs. 37,39–41.

Inference in ACE essentially requires summation operations (Fig. 1a). Hence, ACE models are computationally highly efficient: they can be more than 100 × faster than GAP on CPU cores while achieving the same level of numerical accuracy[39,42]. In contrast, we note that GAP has a high data efficiency (and "learning capability"), enabling efficient collection of initial training datasets, especially at the early stages of fitting (see ref. 43 for an overview of the GAP framework). We recently developed an initial ML potential for elemental tellurium using GAP, and then re-fitted the reference data using ACE, to study crystallisation and melting in Te-based selector devices[44]. Here, we address the structurally and chemically more complex GST system, starting from the existing GST-GAP-22 dataset–which incorporates relevant domain knowledge[31]–and also making use of the ACE framework. However, simply re-fitting an ACE potential using the GST-GAP-22 training data "out of the box" produced unphysical structural motifs in our tests (e.g., atomic clustering and phase segregation), and lost atoms in MD simulations. We believe that this is due to two reasons: first, the hyperparameters used in ACE are complex and can be difficult to tune manually (cf. Supplementary Table 2); second, an ACE model may require more training structures than GAP[45]. The usefulness of ACE potentials in the field of GST materials has been demonstrated recently[38]: the authors focused on the use of a previously proposed indirect-learning approach for ML potentials[46] and built upon the GST-GAP-22 dataset and model[31], reporting efficient ACE potentials which were applied to the $Ge_2Sb_2Te_5$ compound[38].

In the present work, we focus on an alternative route, both optimising the hyperparameters of the model and extending the DFT training dataset (Fig. 1b). The GST-GAP-22 dataset was first re-labelled using DFT with the Perdew–Burke–Ernzerhof (PBE) exchange–correlation functional[47] that is widely used in simulations of PCMs. We then added further AIMD configurations of disordered GST (taken from ref. 31) and fitted initial ACE models to the combined data, using the XPOT software to optimise hyperparameters (Methods)[48–50]. Starting from this well-parameterised ACE model (denoted "iter-0"), we carried out three domain-specific iterations (iter-1 to iter-3) to include melt-quenched disordered structures and intermediate configurations during phase-transition processes (Fig. 1b). These stepwise iterations, acting as a "self-correction" process, provide feedback that enables the potential to correct errors and inaccuracies emerging in its own simulations. We also added small-scale hard-sphere random structures (6–40 atoms) with small atomic distances, generated using the `buildcell` code of ab initio random structure searching (AIRSS)[51,52], for iter-1 to iter-3, to make the ACE models more robust[42]. In addition, we carried out ACE-driven random structure searching (ACE-RSS), akin to previously described GAP-driven RSS[53–55], in two further iterations (iter-4 and iter-5). We refer to our final ACE potential model as "GST-ACE-24". With its training dataset covering multiple GST compositions, we found that our GST-ACE-24 model is chemically transferable along the entire GeTe–$Sb_2Te_3$ tie-line: it can accurately capture different structural properties of various amorphous GST compounds, as validated against AIMD (Methods).

To evaluate the computational efficiency of GST-ACE-24, we performed weak and strong scaling tests on ARCHER2, a high-performance computing system in the UK (Methods and

Supplementary Note 2). We found that, compared to GST-GAP-22, GST-ACE-24 offers more than 400 × higher efficiency on this large-scale CPU architecture (Fig. 1c). Both ACE and GAP showed reasonable scaling behaviour up to 512 nodes (65,536 CPU cores) in strong scaling tests for a structural model of 1 million atoms (Fig. 1d). An efficiency drop-off from the ideal scaling behaviour occurred for the ACE model when handling "small" structures (e.g., 100,000 atoms) on many computing nodes (Supplementary Fig. 1), because ACE is so fast that the inter-processor communication outweighs the computational cost of predicting energies and forces. For example, ≈ 30% of CPU time was used in inter-processor communication when simulating a 100,000-atom structural model on more than 128 nodes. In addition, we found the system-size limit for a total memory of 512 GB to be ≈ 450,000 atoms with GAP, whereas for the same hardware the limit was ≈ 650 million atoms with ACE. Hence, ACE is memory-efficient and enables billion-atom MD simulations (Fig. 1e) with only modest computational resources (e.g., 8 nodes on ARCHER2).

Moreover, ACE can also be used on GPU hardware. We tested device-scale ACE-MD simulations on up to four NVIDIA A100 GPU cards (Methods), and found that the compute-time requirement for ACE-MD on one such GPU card is of the same order of magnitude as that on one 128-core CPU node on ARCHER2 (Supplementary Fig. 2). However, a direct comparison between CPU and GPU is not entirely meaningful due to differences in hardware, parallel computing capabilities, and software-level optimisation for computational tasks. For example, the recursive evaluator developed for ACE–enabled via the `pacemaker` package in LAMMPS and designed to further increase computational efficiency–is currently only compatible with CPU. We also found that ACE's speed compares favourably to state-of-the-art graph-neural-network architectures: our ACE model is about 6 × faster than an equivariant neural-network potential that we directly re-fitted for comparison, using the same training data as for GST-ACE-24 and the MACE architecture[34,56], when testing on an NVIDIA A100 GPU card (Methods). We found the system-size limit for a total memory of 80 GB to be ≈ 92 million atoms with ACE on GPU, smaller than the limit on CPU (≈ 650 million atoms). Hence, while ACE-MD can be run on GPU, its excellent scaling behaviour across multiple CPU nodes and the potential large memory capacity of CPU nodes make it particularly suitable for device-scale MD simulations on existing CPU hardware.

For practical MD simulations, we emphasise the importance of the robustness of ACE models: ACE-MD simulations will fail when atoms are lost due to inaccurately predicted energies and forces. This usually stems from insufficient training data for complex atomic environments with small atomic distances. We designed a protocol to quantify the robustness of ACE models via high-temperature annealing: starting with a hard-sphere random structure of 1000 atoms, the model was annealed at 3000 K for 500 ps. (We note that high-$T$ annealing is part of the melt-quench process to generate amorphous GST, allowing the simulation to visit high-energy configurations; we also note that high-$T$ MD has been used previously for stability tests[57].) We tested the robustness of ACE models on 7 different GST compositions, from GeTe to $Sb_2Te_3$, and performed 10 independent high-$T$ annealing runs for each composition. Despite gradually adding hard-sphere random structures to the training dataset, no successful runs were observed from iter-1 to iter-3: very close interatomic contacts were found in the MD simulations (Fig. 2a), which led to large forces and then lost atoms. However, the inclusion of ACE-RSS structures in the training is concomitant with some successful runs in iter-4 and consistently successful runs in iter-5, producing reasonable high-$T$ liquid GST structures (Fig. 2a).

## Ablation studies for the ACE model

In ML research, "ablation" studies mean gradually removing aspects of a complex model and testing the effect of that on the performance. Here, we report systematic ablation studies for ACE models, with an

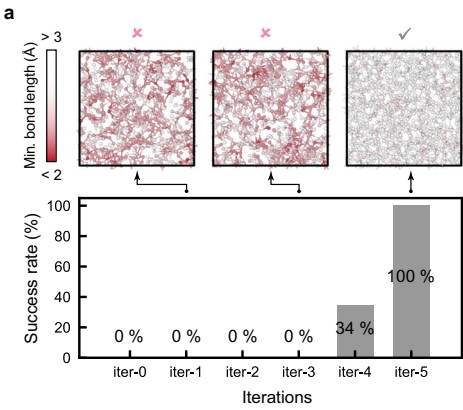

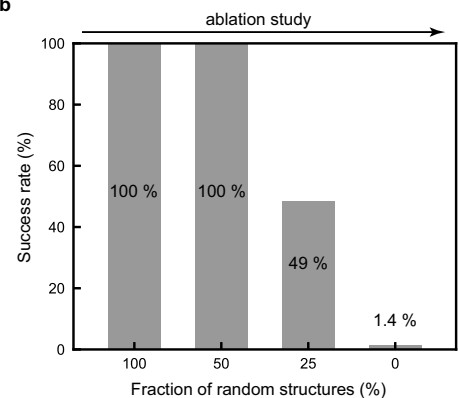

**Fig. 2 | Ablation studies for GST-ACE-24. a** Fraction of successful runs in high-temperature annealing tests for each iteration of the potential. Three structural models from different tests are shown, with atoms colour-coded by minimum bond length: red colour indicates atoms with very short distances to their nearest neighbour, which might lead to lost atoms in molecular-dynamics simulations. **b** An ablation study for the number of random structures present in the training data, evaluated based on the fraction of successful high-$T$ annealing runs. The success rate was obtained across 7 different Ge−Sb−Te compositions, from GeTe to $Sb_2Te_3$, with each composition including 10 independent high-$T$ annealing runs.

aim to more systematically understand the roles of newly added configurations and optimised hyperparameters in ACE models. We first carried out an ablation study for the removal of random structures (including the hard-sphere random and ACE-RSS structures) based on high-temperature annealing tests (Fig. 2b). We observed consistently successful runs when more than half of the random structures were retained. However, removing 75% of them resulted in a marked decrease in stability, and a model where no random structures were present showed almost no successful runs.

We note that removing random structures slightly improved the numerical accuracy in describing domain-relevant structures (e.g., in melt−quench and phase-transition processes), in terms of energies (up to 3 meV atom$^{-1}$) and forces (up to 13 meV Å$^{-1}$) in our ablation tests (Table 1); however, for practical purposes, we do not expect that this small advantage will typically outweigh the risk of losing atoms described above. Hence, in the context of ongoing research on how datasets for ML potentials are best developed[21,58–61], we highlight the key role of such small-scale RSS structures: not only as a starting point for fitting potentials[54,61–64], but also as an effective *post-hoc* correction approach that can add substantial MD stability.

We also carried out ablation studies for the complexity of ACE models. We changed the number of atomic properties, $P$, which controls how the atomic energy is constructed from the local atomic properties in a linear ($P = 1$) or non-linear way ($P \geq 2$). We fitted a linear ACE model and a simpler non-linear one with $P = 2$, and compared both resulting models against GST-ACE-24 ($P = 3$). For the linear model, we found a force RMSE approximately 30 meV Å$^{-1}$ higher than for GST-ACE-24. We also compared ACE models with gradually reduced basis functions (to 1500, 750, and 300, respectively). Although decreasing the number of basis functions increases the computational efficiency (Table 1), fewer functions lead to larger numerical errors. Hence, we argue that our GST-ACE-24 model offers a favourable combination of robustness and accuracy for practical MD simulations.

**Full-cycle operations for cross-point GST memory devices**

With the help of ACE, we are now able to simulate full-cycle operations in cross-point GST devices (Fig. 3a). We first reproduced a non-isothermal melt-quench (RESET) simulation that had previously been demonstrated for cross-point memory, using the GST-GAP-22 potential at the time[31]. As in ref. 31, we used a structural model of $Ge_1Sb_2Te_4$ of $20 \times 20 \times 40$ nm$^3$ (532,980 atoms), which includes a fixed (here, amorphous) slab to prevent unwanted atomic migration across the

periodic cell boundary (Supplementary Note 3)−resembling a thermal barrier in contact with GST in a real device (Fig. 3a).

We used the NVE ensemble (i.e., constant number of particles, volume, and energy) to simulate the RESET process. As shown in Fig. 3b, a 10-ps heating pulse (0.064 pJ) imposed on the model was first simulated by spatially inhomogeneously increasing the kinetic energy of atoms linearly in the direction of the z-axis, corresponding to a large temperature gradient from the bottom (>1000 K) to the top (≈ 300 K) of the 40-nm-long cell. To model the cooling process after removing the heating pulse, the added energy was then gradually removed from each atom over another 40 ps until reaching room temperature. We note that such picosecond-scale data-erasure time is experimentally accessible, evidenced by femtosecond laser experiments[65] and picosecond-scale optical pulses used in an all-optical calculator[66]. The atoms in Fig. 3 are colour-coded based on the smooth overlap of atomic positions (SOAP) kernel similarity[67], which was previously used to quantify per-atom crystallinity for GST[18]. Based on the ACE-MD trajectories, we found that almost all of the structural model turned into amorphous $Ge_1Sb_2Te_4$ after heating and cooling (Supplementary Fig. 3). Technical details of these non-isothermal heating and cooling simulations using NVE are given in Supplementary Note 3 and in our previous work[31]. We note that this RESET simulation using GST-ACE-24 and its evolution of temperature gradients is consistent with previous results using GST-GAP-22 (Supplementary Fig. 3)[31], providing further validation of the present approach.

We next simulated the SET process of the cross-point structural model. Unlike the short, intense RESET pulse (10 ps, 0.064 pJ) that generates a pronounced temperature gradient across the cell, the SET heating pulse has a much longer duration (e.g., tens of nanoseconds) and smaller amplitude, resulting in a lower temperature gradient and smaller fluctuations. Here, we simulated the crystallisation of the device-scale model using the NVT ensemble (i.e., constant number of particles, volume, and temperature). The crystallisation of undoped GST is known to be driven by homogeneous nucleation[68], in which critical nuclei quickly form during a stochastic incubation process[16]. The latter is the bottleneck for crystallisation, which can be bypassed either by applying a low-voltage seeding pre-pulse[69,70] or by doping with a suitable transition-metal element[71–75]. We note that in previous nucleation simulations of GST using AIMD, enhanced sampling methods, e.g., meta-dynamics[76,77] or pre-embedded crystalline seeds[72,78], were employed to accelerate, or circumvent, the formation of critical nuclei in small-scale structural models.

**Table 1 | Ablation studies exploring varying numbers of quantities in ACE models, including: (i) random structures in the training; (ii) atomic properties (see also ref. 45); and (iii) basis functions, based on computed root-mean-square error (RMSE) and relative molecular-dynamics (MD) speed with reference to GST-ACE-24**

|  | Quantities | Energy RMSE (meV atom$^{-1}$) | Force RMSE (meV Å$^{-1}$) | MD speed |
|---|---|---|---|---|
| GST-ACE-24 (reference) |  | 18 | 135 | 1.0 |
| Number of random structures | 50 % | 17 | 129 | 1.0 |
|  | 25 % | 16 | 126 | 1.0 |
|  | 0 % | 15 | 122 | 1.0 |
| Number of atomic properties, $\varphi^{(P)}$ | $P = 2$ | 21 | 142 | 1.0 |
|  | $P = 1$ | 23 | 165 | 1.0 |
| Number of basis functions | 1500 | 22 | 144 | 1.5 |
|  | 750 | 23 | 153 | 2.1 |
|  | 300 | 23 | 168 | 3.1 |

We show computed RMSE values for energy and force predictions on the testing dataset that contains relevant configurations from melt-quench and phase-transition processes. The GST-ACE-24 model developed in this work serves as the reference model in the ablation study, which has 906 random structures (319 hard-sphere random and 587 ACE-driven random-structure searching structures; Supplementary Table 1) in the training dataset, $P = 3$ atomic properties, and 3000 basis functions.

GST-ACE-24 is able to describe nucleation in GST without such additional constraints. We annealed the device-scale structural model of amorphous GST at 600 K for 20 ns, which corresponds to typical electrical pulse durations in GST-based devices[72,79,80]; however, fast crystallisation was observed within several nanoseconds. At 600 K, tens of nucleation centres, with random grain orientations, spontaneously formed after a few nanoseconds. The crystal grains quickly grew at 3 ns, with grain sizes increasing further until 20 ns (Fig. 3c). We analysed this 20-ns SET simulation using the ACE extrapolation grade[81], γ, which allows one to classify atomic configurations into interpolation and extrapolation regimes (Supplementary Fig. 4). Almost all atomic environments fell comfortably within the interpolation regime of our GST-ACE-24 potential, indicating that the nucleation and growth processes are accurately captured by our ACE model. The resulting SET state is a polycrystalline sample of rock-salt-like GST. We counted 277 crystalline grains of different crystal orientations, and the average diameter was ≈ 4.6 nm, consistent with the experimentally measured grain size in GST thin films using in situ transmission electron microscopy[82].

We next simulated a second RESET process of the device-size model. We imposed a 40-ps heating pulse (0.036 pJ) to melt the recrystallised structure. The evolution of temperature profiles is shown in Supplementary Fig. 5. We note that the energy of this heating pulse (0.036 pJ) is smaller than that (0.064 pJ) initially used to erase the initial state of the cell (trigonal layered GST; cf. Fig. 3b); however, this smaller heating pulse still melted the whole structural model. Both the first and the second RESET pulses led to the formation of amorphous GST, with overall similar local structure compared to the results of small-scale, DFT-accessible models (Supplementary Fig. 6). We note that the overall power consumption in these simulations is much lower than that in real devices, because the input power here is directly assigned to specific atoms to increase their kinetic energy. To programme a device experimentally, the thermal energy is generated by Joule heating via electrical pulsing, which involves thermal dissipation and energy loss. Therefore, our ML-driven MD simulations provide the theoretical minimum energy values for RESET operations[31]. Nevertheless, the reduced RESET energy in our simulations implies that a polycrystal, with numerous rock-salt-like crystal grains, is much more easily melted than the stable trigonal phase of GST. We found that the structural disordering primarily occurred at the disordered grain boundaries, similar to the onset of the melting in simulations of re-crystallised, polycrystalline Te[44]. However, our ACE-MD simulations showed that the melting of GST also occurred inside the crystal grains; the latter has been suggested to stem from atomic migration and vacancy diffusion in rock-salt-like crystalline GST[83].

We show the evolution of the potential energy and the fraction of crystal-like atoms during the ACE-MD simulated full-cycle operations in Fig. 3d. These properties provide a quantitative measure of the energetics involved in switching, and reveal the degree of structural ordering at different stages of the device operations. We estimate that the full-cycle simulations (i.e., RESET to SET and back to RESET) using ACE consumed ≈ 770,000 CPU core hours and ≈ 2500 kWh running on ARCHER2 (cf. ref. 84). With more CPU resources available, it is feasible to simulate multiple SET−RESET cycles of GST-based binary memory devices, allowing atomic-scale investigations of structural and compositional variations over repeated full-cycle operations.

## Full-cycle operations for in-memory computing

Beyond their application in data-storage devices, GST alloys have also been used in neuromorphic in-memory computing tasks, which aim to process and store data directly within the same memory cell, thereby avoiding frequent data transfer between conventional memory and processing units[4,5]. In addition to binary ones and zeroes, in-memory computing requires multiple distinct intermediate logic states to represent (near-) continuous weights or values, which are essential for analogue computations (e.g., matrix−vector multiplications). In fact, the electrical-resistance level of GST depends on the ratio of the crystalline to the amorphous volume, making it possible to obtain multiple logic states via appropriate iterative RESET and cumulative SET operations. Such operations can be achieved using small-size bottom electrodes and large programming volumes in mushroom-type devices[8]. As shown in Fig. 4a, given a large programming volume, heating pulses of different amplitudes can thus create mushroom-like active regions with very different crystalline-to-amorphous ratios. Given that the diameter of the bottom electrode can be scaled down to ≈ 3 nm (ref. 85), the dimensions of state-of-the-art mushroom-type devices[4,5] could be further miniaturised, from hundreds to tens of nanometres−providing a broad, tuneable range of cell dimensions for optimisation.

Here, we demonstrate ACE-driven, full-cycle simulations of such partial programming in mushroom-type cells. We simulated a cross-section of GST of $100 \times 40$ nm$^2$, which represents the programming in the middle of a mushroom-type cell (Fig. 4b). We set the thickness of the slab model to 5 nm, corresponding to a quasi-two-dimensional periodic box. In total, this structural model contains 794,808 atoms, much larger than the model size used to describe a mushroom-type geometry in our previous work[31]. The initial configuration is a rock-salt-like crystalline phase of Ge$_1$Sb$_2$Te$_4$, corresponding to an idealised single crystal with cation/vacancy disorder but no grain boundaries. A heat barrier (≈ 6-nm-thick slab of amorphous Ge$_1$Sb$_2$Te$_4$) was added on

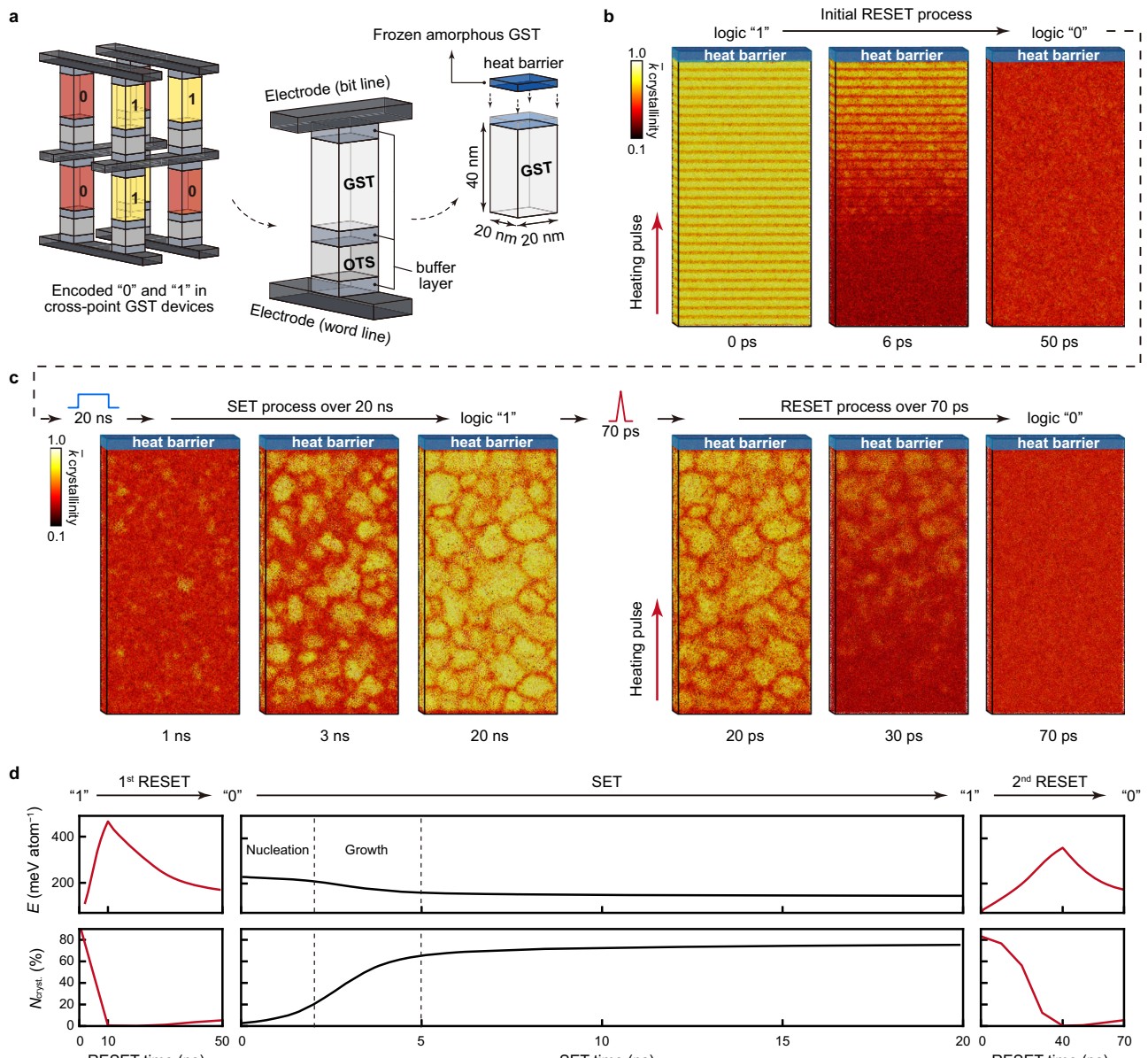

**Fig. 3 | Full-cycle device-scale simulations of cross-point Ge–Sb–Te (GST) devices. a** Schematic of cross-point devices, in which logic "0" and "1" bits are encoded by amorphous and crystalline states of GST. In these devices, GST layers and ovonic threshold switching (OTS) selector layers are sandwiched by buffer layers. A device-size structural model was built as in ref. 31 (20 × 20 × 40 nm³; 532,980 atoms); note that we here use amorphous GST as a heat barrier. **b** The initial RESET operation, simulated similar to ref. 31 but now using the GST-ACE-24 model, starting from layered trigonal Ge₁Sb₂Te₄, triggered by a 10-ps heating pulse (0.064 pJ) from the bottom to the top. After the programming pulse, a 40-ps cooling was performed by removing the added kinetic energy from the structural

model until it reached 300 K. **c** The subsequent SET operation at 600 K, simulated over 20 ns in the NVT ensemble. The resultant recrystallised GST structure contains 277 crystalline grains with an average diameter of ≈ 4.6 nm. A second RESET operation was then simulated, via a 40-ps heating pulse (0.036 pJ) and a 30-ps cooling process (Supplementary Note 3). Colour coding in (**b**–**c**) indicates the smooth overlap of atomic positions (SOAP)-based crystallinity measure, $\bar{k}$ (see ref. 18). **d** Computed potential energy and fraction of crystal-like atoms during the full-cycle simulations. A $\bar{k}$ cut-off of 0.57 was used to separate crystal-like and amorphous-like atoms[18]. Dashed lines indicate the nucleation and growth processes during crystallisation.

the top of the cell, preventing atomic migration across the periodic boundary (Fig. 4b). To simulate programming operations, heating pulses with different magnitudes were applied to regions of different sizes, representing separate logic states (Fig. 4c). We first added a small heating pulse (0.011 pJ) over 100 ps, resulting in a melted programming region with a diameter of ≈ 50 nm. This structural model was then quenched to 300 K over 200 ps by gradually removing kinetic energy from the structural model. We call the resulting intermediate state "logic state I". We note that atoms outside the programming domain remained crystal-like after the heating process, leading to a large crystalline–amorphous interface (Fig. 4c).

We then simulated the crystallisation process for the logic state I at 600 K (Fig. 4 d). Fast crystal growth proceeded at the crystalline–amorphous interface, leading to an evident shrinkage of the disordered-like region. Meanwhile, multiple nuclei were found inside the programming region. The crystalline seeds quickly grew in size, forming a polycrystalline domain. By distinguishing between atoms recrystallised through growth and those through nucleation, we observed a competition between growth-driven and nucleation-driven crystallisation (qualitatively similar to a recent study[86] based on a neural-network potential; see Discussion section for details). In our simulation, the growth-driven crystallisation accounts for

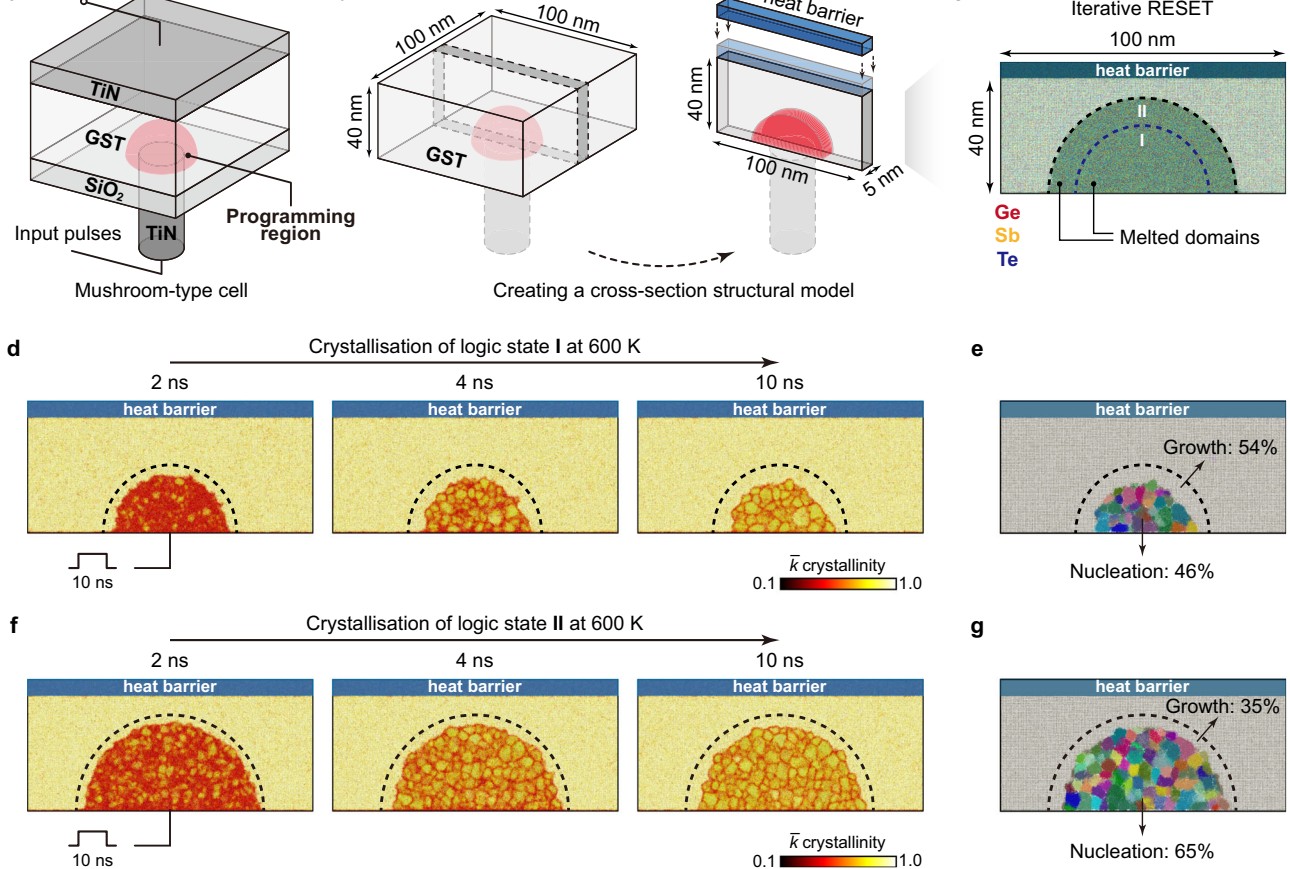

**Fig. 4 | Multiple logic states in a Ge–Sb–Te (GST)-based mushroom-type device.**
**a** Schematic of mushroom-type cells, in which GST layers are sandwiched by top and bottom electrodes (e.g., TiN). **b** A two-dimensional slice model[31] was built (here, $100 \times 40 \times 5\,nm^3$; 794,808 atoms), which represents a cross-section of a mushroom-type cell. **c** Two different intermediate RESET states (viz. logic states I and II) were obtained after different 100-ps heating pulses (0.011 and 0.022 pJ, respectively) and the subsequent 200-ps cooling process. The two melted domains have diameters of ≈ 50 and ≈ 70 nm, respectively, indicated by two different dashed lines. Ge, Sb, and Te atoms are rendered as red, yellow, and blue, respectively. **d** Crystallisation of the intermediate state I at 600 K over 10 ns. **e** Grain-segmentation analysis of the recrystallised intermediate state I. **f–g** As panels (**d–e**), but now for the intermediate state II. Colour-coding in (**d**, **f**) indicates the smooth overlap of atomic positions (SOAP)-based $\bar{k}$ crystallinity (see ref. 18), illustrating the amorphous-like (red) and crystalline-like domains (yellow). The grain analyses shown in panels e and g were carried out using polyhedral template matching[112] and grain-segmentation analyses, as implemented in OVITO[110]. Different colours indicate crystal grains with different orientations. The "growth" region in **e** and **g** highlight the recrystallisation driven by the growth at the crystalline-amorphous interface. The dashed lines in **d** and **e** represent the initial domain of logic state I, and those in **f** and **g** indicate the initial domain of logic state II.

54% of the recrystallised atoms, whereas nucleation contributed 46% (Fig. 4e).

We next added a larger heating pulse (0.022 pJ) to the recrystallised model and cooled it down to 300 K, which created a larger melt-quenched glassy region with a diameter of ≈ 70 nm. We call this intermediate state "logic state II" (Fig. 4c). In its subsequent crystallisation at 600 K (Fig. 4f), the contributions from the growth and nucleation were 35% and 65%, respectively (Fig. 4g). The increased nucleation contribution stems from the dominant nucleation-driven nature of the crystallisation in GST under these conditions. The larger the amorphous region, the more widespread the occurrence of homogeneous nucleation. This finding also implies that the SET speed in GST-based mushroom-type devices at 600 K is almost independent of the size of amorphised regions and the amplitude of the preceding RESET pulse. Rapid homogeneous nucleation is the key to such fast SET operations.

In fact, GST-based in-memory computing devices exhibit considerable resistance noise and time-dependent drift that erodes the precision and consistency of these devices[87,88]. On the one hand, the varied recrystallised morphologies, which contained crystal grains of different orientations (Fig. 4e, g), can be the source of stochasticity in cumulative SET operations, leading to cycle-to-cycle and device-to-device variations. On the other hand, the prominent resistance drift, believed to stem from structural relaxation of amorphous GST (known as ageing), can result in the overlap of two adjacent logic states, causing decoding errors[89]. We show in Supplementary Fig. 7 that our ACE model can well describe the degree of local bond-length asymmetry, sometimes referred to as Peierls distortions, of amorphous GST —a quantitative structural fingerprint of the ageing process[90]. Hence, our ACE model can simulate both stochastic recrystallisation and aged amorphous structures of mushroom-type devices, which provides atomic-scale insights into the programming mechanisms of GST-based mushroom-type devices for in-memory computing tasks.

## Discussion

Our ultrafast and chemically transferable ACE potential for GST alloys can serve as a powerful "off-the-shelf" simulation tool with quantum-mechanical accuracy. Its computational efficiency enables full-cycle simulations (multiple RESET to SET operations) of different device architectures at extensive length scales (tens of nanometres) and time scales (tens of nanoseconds). We expect that our ACE model can provide atomic-scale insights into realistic programming conditions of GST-based electronics, including repeated switching for binary memory applications, as well as cumulative SET and iterative RESET

processes for neuromorphic in-memory computing. In the latter case, larger device geometries than in our current proof-of-concept simulation (Fig. 4) could make it possible to more finely tailor the amorphous-to-crystalline volume ratio to accommodate more resistance states. Simulating complex in-memory operations at the atomic scale could provide a more in-depth understanding of phase-change neuromorphic computing, and such simulations would benefit from fast and efficient ACE models. Moreover, our ACE model could also offer useful atomic-scale perspectives for GST-based waveguide memories and other emerging optical technologies[91–94]. Unlike compact electronic devices, waveguide devices typically feature less confined geometries and require the use of the NPT ensemble (i.e., constant number of particles, pressure, and temperature) to address potential volume changes during switching processes. We note that ACE-MD simulations are well-suited to these scenarios, as they can handle the required time (tens of nanoseconds) and length scales (tens of millions of atoms).

We note that the atomistic modelling of PCMs on large length scales is gaining increasing interest in the community. From a technical perspective, the indirectly-learned ML potential of ref. 38 already illustrated the usefulness of ACE in this domain: the authors reported the simulation of a ≈ 1-million-atom bulk $Ge_2Sb_2Te_5$ structure over 1 ns on combined CPU and GPU architectures, as well as the repeated switching of a ≈ 100,000-atom bulk structure[38]. The scaling tests described in ref. 38 are qualitatively consistent with our tests on the ARCHER2 high-performance computing system (Fig. 1c–e) where applicable, although (as the authors also note) the details will depend on the specific hardware. In terms of simulation cells and protocols reflecting PCM device geometries, a recent study described the use of a neural-network ML potential for $Ge_2Sb_2Te_5$ (from ref. 25) and multiple GPU cards to perform large-scale simulations (≈ 2.8 million atoms) over several nanoseconds[86]. A structural model was created by embedding an amorphous dome in a crystalline matrix to represent a mushroom-type device, and multiple thermostats were used to simulate SET operations[86]; a competition between nucleation and growth from the interface was identified[86], which is qualitatively similar to Fig. 4. In our present work, we have combined a carefully optimised ACE potential that makes efficient use of CPU resources with advanced simulation protocols for cell geometries and programming conditions that are relevant to both cross-point and mushroom-type GST devices.

Although we employed a fixed amorphous GST slab as a heat barrier (cf. Figs. 3–4) to approximate the impact of interfaces present in real devices (such as TiN or $SiO_2$ contacts), incorporating interface effects of the surrounding materials in a realistic way is an important future step—and is expected to be technically feasible, especially given the availability of relevant previously published training datasets, e.g., for the full Si−O binary system[45]. A key challenge in this extension will likely be to construct representative configurations that capture complex interface interactions involving four or more elements. To address this challenge, we highlight the use of GAP in rapidly sampling diverse chemical and structural space with minimal prior knowledge[61,95], thereby facilitating the construction of an initial training dataset for interfaces. As demonstrated in the present work, such GAP-based datasets can be directly fed into ACE training, followed by further domain-specific iterations. In addition, uncertainty-based sampling for active learning can be used to obtain a more comprehensive training dataset[81].

Looking back on the discussion of PCM modelling at the beginning of this paper, we note that ML-driven simulation methods have now been established in the field, allowing for wide-ranging simulation studies of functional materials, and increasingly becoming of relevance to experimental work and practical applications. Our study has exemplified this advance for the field of electronic memories and neuromorphic in-memory computing, and other atomistic ML models have been developed for a wide range of applications across different disciplines: recently published ML potentials have been used in the search for new stable inorganic crystals (e.g., for layered materials and solid-electrolyte candidates)[96], in the prediction of supercritical behaviour in high-pressure liquid hydrogen, relevant to the structure and evolution of giant planets[97], or in biomolecular-dynamics simulations of protein-folding processes and their thermodynamics[98]. The relevance of ML-driven simulations in the computational design of amorphous materials—PCMs and many others—has been pointed out in ref. 99. A very recent preprint discusses the role of ML potentials for device-scale modelling in a wider perspective[100]. We expect that our present work will stimulate the further development of efficient ML potentials for exploring structurally and chemically more complex PCM systems and devices, and provide a key approach for investigating scientific questions related to memory and computing applications.

## Methods

### The GST-ACE-24 potential

All ACE models shown in the present work were fitted using pace-maker (version 0.2.7; ref. 40); their optimisation was carried out with XPOT (version 1.1.0; refs. 48,50). The extension of XPOT to ACE specifically, and the physical role of relevant hyperparameters, has been discussed in our more technical study in ref. 49. The latter includes investigations of ACE models for silicon and the binary compound $Sb_2Te_3$ and provides a basis for the present work.

Using XPOT, we optimised 4 hyperparameters (cf. Supplementary Table 2) based on the iter-0 dataset and performed 32 fitting iterations (cf. Fig. 1b). To guide the target of the XPOT optimisation, we defined a testing dataset consisting of conventional disordered structures (≈ 200 atoms each) and intermediate configurations during phase transitions (1008 atoms each). These two types of structures were taken from AIMD simulations reported in ref. 31 and ref. 18, respectively. This testing dataset was also used in the computation of RMSE values shown in Table 1. In the XPOT optimisations, we first performed 8 exploratory fits using a Hammersley sequence to sample hyperparameters. Next, Bayesian Optimisation (BO) was used to optimise the hyperparameters over the remaining iterations. After XPOT optimisation, we "upfitted" the best potential (with an increased relative weighting of the energy; see ref. 49). In fact, after iter-3, we performed another XPOT run to determine whether the model required further hyperparameter optimisation based on the newly added configurations (i.e., those from iter-1 to iter-3). However, we found no notable improvements in accuracy on the testing dataset, and therefore continued with the existing hyperparameters, as optimised on the iter-0 dataset. We note that the hyperparameters determined here for GST-ACE-24 were also used in fitting a separate ACE model for elemental tellurium, which is described in ref. 44.

The final potential model combines linear and nonlinear embeddings of the atomic neighbour environments over 3000 basis functions and uses a radial cut-off of 8 Å. Training structures were weighted, based on their configuration types. Crystalline structures, melt−quench structures from AIMD, and RSS structures were given custom weightings to guide model accuracy in these regions. The model was fitted using an NVIDIA A100 GPU.

We note that a positive core-repulsion term can be included in ACE models to stop unphysical energies and forces from being produced at short atomic distances and to correct the core-repulsion behaviour, which can mitigate simulation issues with "lost" atoms[40]; such an approach has been taken for the ACE models of ref. 38. By contrast, adding high-energy, small-scale random structures helps to explore a diverse configurational space (see, e.g., refs. 51,55). In particular, such additional configurations improve the ability of potentials to describe two-body interactions at unusual distances, preventing the potentials from predicting the formation of clusters which, when evaluated with DFT, were found to be energetically unfavourable. We

note that the addition of training data to represent short interatomic distances, "rather than relying on the core repulsion completely", has been suggested by the `pacemaker` developers (see ref. 101). Our GST-ACE-24 model does not use a separate core-repulsion term.

## Validation

We computed different structural properties of various amorphous GST compounds along the GeTe−$Sb_2Te_3$ compositional tie-line, such as radial and angular distribution functions (Supplementary Fig. 7), and found that the predictions of our GST-ACE-24 model agreed very well with the AIMD data of ref. 31. Also, our ACE model faithfully reproduced the fraction of homopolar bonds and tetrahedral motifs, as well as the degree of local bond-length asymmetry (Supplementary Fig. 7), which are important structural factors that have been discussed in the context of ageing phenomena in the amorphous phase[90]. These structural validations demonstrate that our new ACE potential is both structurally and chemically transferable and can accurately describe disordered GST structures across various compositions, consistent with results for the GST-GAP-22 model[31].

## Computational performance

A key point in the present study is how ACE allows for ultra-fast device-scale simulations on a CPU-based high-performance computing system, without requiring GPU hardware at runtime. In addition to the simple and fast summation operations performed in the construction and inference of the ACE model, a recursive evaluation algorithm is used to construct the basis functions, reducing the number of arithmetic operations, and thus improving numerical efficiency[39]. We measured the performance of our GST-ACE-24 model by comparing against the published GST-GAP-22 potential[31] on the CPU cores of the ARCHER2 system. The compute nodes each have 128 CPU cores, and the memory per node is either 256 GB (standard nodes) or 512 GB (high-memory nodes); see ref. 102 for details. The comparison between ACE and GAP is shown in Fig. 1c−e.

We note that ACE also supports multi-GPU computation: our test on a 1-million-atom structural model showed good scalability of ACE-MD simulations running on up to four NVIDIA A100 GPUs with 80 GB of memory each (Supplementary Fig. 2). In addition, we compared the computational efficiency of GST-ACE-24 with a directly re-fitted equivariant neural-network potential, based on the MACE architecture[34,56], on a GPU. This directly re-fitted MACE model used the same training dataset of GST-ACE-24. We performed the tests for GST-ACE-24 and the MACE model on an NVIDIA A100 GPU with 80 GB of memory, using a 10,000-atom structural model. The computational efficiency of GST-ACE-24 in this setting was ≈ 2 million MD steps per day, whereas the computational efficiency of the MACE model was ≈ 335,000 MD steps per day.

## DFT computations

The AIMD data used for the fitting process (Fig. 1b) and the validation (Supplementary Fig. 7) of our ACE model were taken from our previous work (ref. 31). These AIMD simulations had been carried out using the "second-generation" Car−Parrinello scheme, as implemented in the Quickstep code of CP2K (version 2023.1)[103], a combination of Gaussian-type and plane-wave basis sets, scalar-relativistic Goedecker pseudopotentials[104], and the Perdew−Burke−Ernzerhof (PBE) functional[47]. Details of the AIMD simulations may be found in ref. 31.

To label the reference dataset, we computed the per-structure energies and per-atom forces by performing single-point DFT computations using the Vienna Ab initio Simulation Package (VASP; version 5.4.4)[105,106] with projector augmented-wave (PAW) pseudopotentials[107,108]. We used a 600 eV cut-off for plane waves and an energy tolerance of $10^{-7}$ eV per cell for SCF convergence. An automatically generated $k$-point grid with a maximum spacing of 0.2 Å$^{-1}$ was used to sample reciprocal space.

## Molecular-dynamics simulations

MD simulations were carried out with the GST-GAP-22 (ref. 31) and GST-ACE-24 ML potential models, using LAMMPS (version 15 Jun 2023)[109], with interfaces to QUIP and `pacemaker`, respectively. The canonical ensemble (NVT) and the microcanonical ensemble (NVE) were used in this work. A Langevin thermostat was used to control the temperature in the NVT simulations. We simulated non-isothermal heating processes in the NVE ensemble. Additional energy was added to the kinetic energy of the atoms in the programming regions (Supplementary Note 3), with a timestep of 2 ps. The timestep for all ML-driven MD simulations was 2 fs. Structures were visualised using OVITO[110].

## Reporting summary

Further information on research design is available in the Nature Portfolio Reporting Summary linked to this article.

## Data availability

The raw data for the figures presented in the main text and Supplementary Information have been provided in a Source Data file with this paper. The rest of the data supporting the present study, including the potential parameters, fitting data and structural models shown in Figs. 3−4, are publicly available via Zenodo at https://doi.org/10.5281/zenodo.14755074 (ref. 111). Source data are provided with this paper.

## Code availability

The XPOT software used for hyperparameter optimisation is available at https://github.com/dft-dutoit/XPOT under the GPL-2.0 licence. A copy has been deposited in Zenodo and is available at https://doi.org/10.5281/zenodo.15853809 (ref. 50). Other software packages were used as provided by their respective authors.

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

## Acknowledgements

Y.Z. acknowledges a China Scholarship Council-University of Oxford scholarship. S.R.E. acknowledges the Leverhulme Trust (UK) for a Fellowship. W.Z. thanks support by the National Key Research and Development Programme of China (2023YFB4404500), the National Natural Science Foundation of China (62374131), the Computing Centre in Xi'an and the International Joint Laboratory for Micro/Nano Manufacturing and Measurement Technologies of XJTU. This work was supported by UK Research and Innovation [grant number EP/X016188/1]. We are grateful for computational support from the UK national high-performance computing service, ARCHER**2**, for which access was obtained via the UKCP consortium and funded by EPSRC grant ref EP/X035891/**1**, as well as through a separate EPSRC Access to High-Performance Computing award.

## Author contributions

Y.Z., W.Z., and V.L.D. designed the study. Y.Z. and D.F.T.d.T. parameterised the ACE potential models. D.F.T.d.T. studied the role of hyperparameters and provided technical advice. Y.Z. carried out the large-scale molecular-dynamics simulations and visualised the results. All authors (Y.Z., D.F.T.d.T., S.R.E., W.Z., and V.L.D.) contributed to discussions and to the writing of the paper.

## Competing interests

The authors declare no competing interests.
