## [Transparent Peer Review file · Nature Communications]

Full-cycle device-scale simulations of memory materials with a tailored atomic-cluster-expansion potential

Corresponding Author: Professor Volker Deringer

Version 0:

Reviewer comments:

Reviewer #1

(Remarks to the Author)

The authors have reported device-scale atomistic simulations of phase-change materials. They have largely improved the computational efficiency of the machine-learned potential that can directly cover the crystallization process in devices, which requires much longer simulation time of amorphization. I regard this work as a significant step forward in atomistic simulations of phase-change materials, which can be considered for publication at Nat Commun. I have some minor suggestions for the authors to consider:

1. Since phase-change materials have also been exploited for optical applications which hold even larger spatial size, I would suggest the authors to include some discussions about possible simulations of e.g. photonic waveguide memory devices as an outlook.
2. If the ACE framework is so efficient in atomistic simulations, is it still necessary to develop the GAP potential for a given material? I noticed that authors also did GAP simulations first and then the ACE simulations for elemental Te in Ref. 44. Some additional discussions in the last part of the manuscript could be very useful to the readers who will study their own materials systems.
3. Considering that the RESET and SET processes in Fig. 3, it looks like the crystallization process can be finished within 5 ns and the amorphization is even faster of 70 ps. Are there any experimental results reported consistent with the speeds simulated in this work?

Reviewer #2

(Remarks to the Author)

The study shows a full-cycle phase change memory device-scale simulations using ACE-based machine learning interatomic potential. The authors provided detailed information on the training strategy and process of the potential, and made comparisons between ACE-based potential and GAP-based potential. It is a well-structured piece of work that makes contributions to this field. Here are some questions that I would like the authors to address.

- 1) The training process of the potential is separated into five steps (iter1 – iter5). What are the benefits of this strategy? Will there be any differences in the performance of the ACE-potential if iter1-3 and iter4-5 are merged into two separate single-iteration training steps?
- 2) The study compares the efficiency of CPU-based and GPU-based computation, testing on one Nvidia A100 GPU. While the authors address that the memory limit of a single GPU restricts the number of atoms in the device-scale simulations, I wonder that whether ACE-based potential supports multi-GPU system? I hope the authors can make a comparison between computational cost and speed of device-scale simulations on a same GST model using GPU and CPU respectively. As is well known, the performance and efficiency of GPU is much higher than CPU.
- 3) There are several large-scale amorphous models in this work. I hope the authors can make detailed structural analysis of these models. Are there any special medium-range or long-range structural features (“fingerprints”) in amorphous GST that can be observed or concluded? As is widely known, only short-range structural information of amorphous GST is clear due to the restrictions of previous DFT-calculated small models.
- 4) According to fig4(a), the structure of the device also includes TiN and SiO₂ other than GST. However, the device-scale

model used for ACE-based computation only composed of free GST atoms and fixed GST atoms. Is this model sufficient to represent the behaviors of actual devices? Why these kinds of interfaces or contacts are not included in the dataset to obtain a more general potential? I hope the authors can make thorough explanation about this.

Reviewer #3

(Remarks to the Author)

The manuscript presents a significant advancement in the atomistic simulation of phase-change materials (PCM) for memory technology applications. Following a systematic approach to constructing and extending the training dataset, the authors successfully developed a computationally efficient and robust machine-learned interatomic potential based on the Atomic Cluster Expansion (ACE) framework for the Ge-Sb-Te (GST) system. This potential enabled unprecedented atomistic simulations of the GST PCM system at timescales and system sizes relevant to real devices, or, as the authors state, simulations of "the real thing." The simulation results, paired with an extensive study of the ACE model's scalability, demonstrate that the applied combination of techniques has the potential to significantly impact the field by enabling more detailed and realistic atomistic investigations of PCM device operation and performance.

While the manuscript is already of high quality and suitable for publication as it is, I have two minor points for the authors to address:

- The ACE model used in this work has the capability to estimate the uncertainty of its predictions. Specifically, for a given atomic configuration, the model can indicate whether its prediction falls within an extrapolation or interpolation regime. Did the authors use this feature during simulations or during the preparation of the training dataset? If not, what was the rationale? While the authors employed a justified approach for selecting relevant structures for the training dataset, uncertainty indication provided by the ACE model could further facilitate this process. For example, it could identify configurations that are furthest from previously observed data, thereby improving the efficiency of structural space exploration. And overall, uncertainty indication provides an additional mechanism to verify the model's transferability.
- After simulating the re-crystallization of GST (the set process), the authors report an average grain size of approximately 4.6 nm. Is there a reference available (e.g., experimental measurement) to which this result can be compared? Comparing structural parameters could help establish a closer connection between simulation conditions and real material behavior.

Reviewer #4

(Remarks to the Author)

Version 1:

Reviewer comments:

Reviewer #1

(Remarks to the Author)

Thank you to the authors for their detailed response. I am satisfied with the response and the revisions made to the manuscript, and I would therefore like to recommend its acceptance.

(Remarks on code availability)

Reviewer #2

(Remarks to the Author)

The manuscript has been well revised with all the problems addressed. The study provides valuable insights into the atomistic simulation of phase-change memory materials. I believe it is suitable for publication in Nature Communications.

(Remarks on code availability)

The authors provide the method for fitting the machine learning potential, along with several examples. The code is well-organized and comes with detailed instructions.

Reviewer #3

(Remarks to the Author)

The reviewer thanks the authors for responding to all points raised in the report. The reviewer has no further requests for changes or clarifications.

(Remarks on code availability)

Response to Reviewers – Manuscript NCOMMS-25-09245

We thank all reviewers for their helpful and constructive comments. We are pleased to read that they judge our work to be “*a significant step forward in atomistic simulations of phase-change materials*” (Reviewer #1) and to have “*the potential to significantly impact the field*” (Reviewer #3), respectively. We also appreciate the reviewers’ questions and suggestions, which have helped us to further strengthen our manuscript.

We provide a summary of changes here, followed by a detailed reply starting on page R2. In brief, the most important aspects are:

- We now make a more direct comparison between our simulations and experimental data, citing additional studies. Specifically, we addressed the questions regarding practical RESET/SET operation timescales (Reviewer #1) and grain sizes observed in recrystallised GST cells (Reviewer #3).
- We added new data and analyses: following the suggestions by Reviewer #2, we tested ACE-MD simulations on multiple GPUs and performed structural analyses on the device-scale amorphous structures. Moreover, we quantified the uncertainty of our ACE model’s predictions during the crystallisation process, providing evidence that our domain-specific iterations have yielded a comprehensive training dataset.
- In response to Reviewer #2’s comment about TiN and SiO₂ interfaces, we now highlight the practical importance and methodological feasibility of incorporating these interfaces in future studies. In the revised Discussion section, we draw a perspective for developing ACE models capable of handling complex interfaces, thus enabling even more realistic device simulations.

We note that we corrected a unit conversion error in Fig. 1c. This correction affects the absolute y-axis values but does not change the relative speed-up of ACE as compared to GAP. We also corrected a labelling error on the last bar of Fig. 2b.

Starting on the next page, we quote the reviewers’ reports in full (in *italics*), and our point-by-point response is interspersed in **blue**. Action taken is described in **red**.

Reviewer #1

The authors have reported device-scale atomistic simulations of phase-change materials. They have largely improved the computational efficiency of the machine-learned potential that can directly cover the crystallization process in devices, which requires much longer simulation time of amorphization. I regard this work as a significant step forward in atomistic simulations of phase-change materials, which can be considered for publication at Nat Commun.

I have some minor suggestions for the authors to consider:

Response: We thank the reviewer for their useful suggestions for improvement. We address these point by point below.

1. *Since phase-change materials have also been exploited for optical applications which hold even larger spatial size, I would suggest the authors to include some discussions about possible simulations of e.g. photonic waveguide memory devices as an outlook.*

Response: Indeed, GST-based phase-change materials are becoming increasingly important for optical applications, including photonic waveguide memories and other emerging technologies (see Refs. R1–R4). We note that ACE-MD simulations are well-suited for the time and length scales of these scenarios. For example, a typical GST waveguide device has dimensions of $\approx 400 \times 400 \times 10 \text{ nm}^3$, which corresponds to ≈ 53 million atoms^{R5}. As illustrated by our computational efficiency tests (cf. Fig. 2 in the main text), MD simulations of 50 million atoms over tens of nanoseconds remains computationally feasible with ACE-MD.

Action taken: We added the following text in the revised manuscript (p. 19), together with relevant references (Refs. R1–R4), discussing potential future applications of ACE-MD simulations for modelling PCM-based optical devices.

“Moreover, our ACE model could also offer useful atomic-scale perspectives for GST-based waveguide memories and other emerging optical technologies^{92–95}. Unlike compact electronic devices, waveguide devices typically feature less confined geometries and require the use of the NPT ensemble (i.e., constant number of particles, pressure, and temperature) to address potential volume changes during switching processes. We note that ACE-MD simulations are well-suited to these scenarios, as they can handle the required time (tens of nanoseconds) and length scales (tens of millions of atoms).”

2. *If the ACE framework is so efficient in atomistic simulations, is it still necessary to develop the GAP potential for a given material? I noticed that authors also did GAP simulations first and then the ACE simulations for elemental Te in Ref. 44. Some additional discussions in the last part of the manuscript could be very useful to the readers who will study their own materials systems.*

Response: While ACE potentials can indeed be fitted from scratch—as shown for, say, magnesium^{R6}, iron oxides^{R7}, and water^{R8}—they typically require larger initial training datasets as compared to GAP. Using an existing and reliable reference dataset as a starting point can help in fitting ACE models^{R9–R11}. We have shown that datasets generated in the process of GAP fitting can be directly re-used to fit robust and fast ACE models for tellurium^{R12} and Sb₂Te₃^{R13}, both of which are similar but chemically simpler material systems compared to GST. In contrast, our first attempt to directly refit a robust ACE potential using the previous GST-GAP-22 dataset failed for the structurally and chemically complex GST, and thus additional iterations were carried out to improve our ACE potential model.

In fact, due to their data efficiency (and “learning capability”)^{R14}, GAP potentials can be more robust than ACE, at very early stages of fitting, to explore the target potential energy surface, and avoid producing unphysical configurations for initial training datasets. Our recent works showcase the usefulness of GAP in rapidly sampling diverse chemical and structural space with very little prior knowledge required. For example, we have introduced automated workflows for structural exploration employing random-structure searching combined with GAP^{R15}.

In summary, GAP and ACE potentials each have their specific strengths: GAP is useful at early exploration and quick collection of initial training datasets, whereas ACE is preferable for large-scale simulations once a robust initial dataset is available. Hence, using GAP-based exploration to first establish an initial dataset might facilitate the development of an ACE potential.

Action taken: We added additional discussions in the revised manuscript, together with relevant references, to clarify the reason for using an existing GAP reference dataset (p. 4) and to highlight the use of GAP-based exploration for complex PCM systems (p. 20).

3. *Considering that the RESET and SET processes in Fig. 3, it looks like the crystallization process can be finished within 5 ns and the amorphization is even faster at 70 ps. Are there any experimental results reported consistent with the speeds simulated in this work?*

Response: Indeed, experimental studies have demonstrated rapid crystallisation and amorphisation in GST-based devices. For the SET process, earlier experimental reports on mushroom-type GST devices have shown that complete crystallisation (SET) can be triggered by electrical pulses as short as 10–20 ns^{R16–R18}. We note, however, that the reported electrical pulses represent the total pulse duration rather than the intrinsic crystallisation time of GST itself. Consistent with these experimental findings, our simulations of the 20-ns SET pulse in cross-point devices (Fig. 3 in the main text) indicate crystallisation occurring on the order of several nanoseconds.

For the RESET operation, we note that the melting of GST can be achieved in 5–10 ps, as evidenced by femtosecond laser experiments^{R19}. Moreover, experiments have shown picosecond-scale optical pulses (several to tens of picoseconds) used in an all-optical calculator for fundamental arithmetic operations^{R20}.

Action taken: We added new discussions, together with relevant experimental references, to compare the RESET and SET timescales of our simulations and experiments (pp. 11–12).

Reviewer #2

The study shows a full-cycle phase change memory device-scale simulations using ACE-based machine learning interatomic potential. The authors provided detailed information on the training strategy and process of the potential, and made comparisons between ACE-based potential and GAP-based potential. It is a well-structured piece of work that makes contributions to this field.

Here are some questions that I would like the authors to address.

Response: We thank the reviewer for their positive comments. We address their questions point by point below.

1. *The training process of the potential is separated into five steps (iter1 – iter5). What are the benefits of this strategy? Will there be any differences in the performance of the ACE-potential if iter1-3 and iter4-5 are merged into two separate single-iteration training steps?*

Response: The iterative training approach used in this work helps progressively improve the potential by systematically capturing diverse structural motifs relevant to phase transitions of GST. Each iteration acts as a stepwise refinement in what could be called a “self-correction” process, whereby the potential learns from and corrects errors (e.g., fictitious local minima) or inaccuracies (due to insufficient sampling) that emerge during previous iterative training steps. We typically continue the iterative process until the potential becomes capable of specific, difficult simulation tasks (e.g., modelling of SET and RESET operations in this work).

We think that merging multiple iterative training cycles (e.g., iter1–3 and iter4–5) into fewer iterations or even a single one might risk losing the feedback loop provided by the above-mentioned “self-correction” process. Without iterative feedback, a single-iteration training step might fail to identify and correct errors arising during simulations. Hence, we make use of iterative training to produce a converged, high-quality training dataset; once this final dataset is obtained, it can be directly used for fitting or adapted in other machine-learning frameworks, allowing future users to bypass the iterative process and benefit from the finalised dataset.

Action taken: We added the following sentence in the revised manuscript (p. 5) to highlight the importance of the stepwise iterative training protocol.

“These stepwise iterations, acting as a ‘self-correction’ process, provide feedback that enables the potential to correct errors and inaccuracies emerging in its own simulations.”

2. *The study compares the efficiency of CPU-based and GPU-based computation, testing on one Nvidia A100 GPU. While the authors address that the memory limit of a single GPU restricts the*

number of atoms in the device-scale simulations, I wonder whether ACE-based potential supports multi-GPU systems? I hope the authors can make a comparison between computational cost and speed of device-scale simulations on the same GST model using GPU and CPU respectively. As is well known, the performance and efficiency of GPU is much higher than CPU.

Response: We thank the reviewer for this constructive suggestion. We note that ACE potentials fully support multi-GPU computation. In response to the reviewer's request, we tested the scalability of ACE-MD simulations on up to four A100 GPUs on the same server. We compared the speed-up of ACE-MD simulations of 1-million atoms running on multiple A100 GPU cards or ARCHER2 CPU nodes of the same number. We found that both CPU and GPU implementations show good scaling behaviour, and that the compute-time requirement for ACE-MD on one such GPU card is of the same order of magnitude as that on one 128-core CPU node on ARCHER2. However, we note that a direct comparison between a CPU node and a GPU card is not entirely meaningful due to differences in hardware, parallel computing capabilities, and software-level optimisation for computational tasks. For example, the recursive evaluator developed for ACE—enabled via the `pacemaker` package in LAMMPS and designed to further increase computational efficiency—is currently only compatible with CPUs. We also note that our GPU scaling tests are performed on a single node. While it is possible to parallelise across multiple nodes, quantifying multi-node GPU performance remains to be completed in future work.

In fact, we find the benefits of using GPUs to be most pronounced during model fitting. With the rapid advancement in GPU computational performance and an improved GPU parallelisation for larger simulation systems at the software level, we envision that GPU-based ACE-MD simulations will become increasingly useful for device-scale modelling in the future.

Action taken: We now mention the multi-GPU support using the LAMMPS ML-PACE package in the revised manuscript (p. 7 and p. 23). In addition, we included a new Supplementary Figure 2 (p. S8) to compare the computational efficiency of ACE-MD simulations running on multiple GPU cards and CPU nodes.

3. *There are several large-scale amorphous models in this work. I hope the authors can make detailed structural analysis of these models. Are there any special medium-range or long-range structural features (“fingerprints”) in amorphous GST that can be observed or concluded? As is widely known, only short-range structural information of amorphous GST is clear due to the restrictions of previous DFT-calculated small models.*

Response: We have now performed additional structural analyses of two device-scale amorphous GST models (obtained after the first and second RESET process, respectively) and compared the results to those for a smaller (364 atoms), “DFT-accessible” structural model (Fig. R1a). As shown in Fig. R1b–c, we computed the radial distribution function (RDF) and angular

distribution function (ADF) for the structural models. They show good agreement, indicating generally consistent short-range structural order across the vastly different model sizes.

To further investigate medium- and long-range structural “fingerprints”, we performed ring analyses on both the small-scale and device-scale models. The results reveal overall similar ring-size distributions, with four-membered rings dominating in both cases. In the small model, five-membered rings are slightly more abundant than six-membered ones, whereas in the device-scale model, there are more six- than five-membered rings. Importantly, both models exhibit a substantial number of large, complex rings (ring length ≥ 8), indicative of the medium-range structural ordering in amorphous GST.

Action taken: We added the following text in the revised manuscript (p. 14), and included the new structural analyses (RDF, ADF, and ring-size distributions) as new Supplementary Figure 6 (p. S12).

“Both the first and the second RESET pulses led to the formation of amorphous GST, with overall similar local structure compared to the results of small-scale, DFT-accessible models (Supplementary Fig. 6).”

Figure R1 (included as new Supplementary Fig. 6). Structural analyses of amorphous GST models with different sizes. (a) Snapshots of amorphous GST, including a small-scale model (364 atoms) that was generated via a melt-quench process using ACE-MD, and a device-scale model (532,980 atoms). (b) Computed radial distribution function (RDF) values, including analysis for device-scale structural models obtained after the first and second RESET pulse, respectively (cf. Fig. 3 in the main text). Only the central volumes of the device-scale models ($20 \times 20 \times 20 \text{ nm}^3$) were used for structural analyses. (c) Same but for computed angular distribution function (ADF) values. (d) Same but for the distribution of ring sizes.

4. According to fig4(a), the structure of the device also includes TiN and SiO₂ other than GST. However, the device-scale model used for ACE-based computation only composed of free GST atoms and fixed GST atoms. Is this model sufficient to represent actual devices? Why these kinds of interfaces or contacts are not included in the dataset to obtain a more general potential? I hope the authors can make thorough explanation about this.

Response: We thank the reviewer for raising this important question. Our current work focuses on the atomic-scale switching dynamics within the GST cell itself. The fixed amorphous GST slab at the top boundary of our simulation serves as a simplified but effective heat barrier, enabling thermal constraints and partially mimicking the impact of interfaces present in real devices (such as TiN or SiO₂ contacts). Although simplified, this configuration enables us to gain meaningful atomic-scale insights into the switching mechanisms and internal structural evolution of the GST active region—key phenomena that are central to understanding device operations.

From a technical perspective, introducing additional chemical elements increases the computational cost of training and running the potential, and so incorporating additional interfaces (such as TiN and SiO₂ contacts) into the current GST potential would substantially increase computational costs as well as the manual efforts required to develop a more general ACE potential. However, we fully acknowledge that extending the ACE potential to model complex interfacial structures is both valuable and technically achievable. In fact, we have previously developed an ACE potential for the full Si–O binary system, capable of accurately describing the high-pressure phases, surfaces, and nanoscale heterogeneity of Si–O systems.

Hence, a next step for future research could be to integrate our current GST dataset with existing or newly constructed datasets for other relevant materials, thus enabling the training of a comprehensive multi-element potential (e.g., GST–Si–O–Ti–N). A key challenge in this regard will be the generation of representative configurations that capture the complex interface interactions (e.g., GST–SiO₂ and GST–TiN), and we argue that building and carefully validating such a model would be far beyond the scope of the present revision. In the revised manuscript, we therefore identify this topic as an avenue for future research.

Action taken: We added a new paragraph in the Discussion section (p. 20) and now clearly mention the approximation made in using a GST slab as a simplified heat barrier. In addition, we now outline future directions for extending our approach to more complex multi-element potentials and the modelling of interfaces (p. 20).

“Although we employed a fixed amorphous GST slab as a heat barrier (cf. Figs. 3–4) to approximate the impact of interfaces present in real devices (such as TiN or SiO₂ contacts), incorporating interface effects of the surrounding materials in a realistic way is an important future step—and is expected to be technically feasible, especially given the availability of

relevant previously published training datasets, e.g., for the full Si–O binary system⁴⁵. A key challenge in this extension will likely be to construct representative configurations that capture complex interface interactions involving four or more elements. To address this challenge, we highlight the use of GAP in rapidly sampling diverse chemical and structural space with minimal prior knowledge^{60,96}, thereby facilitating the construction of an initial training dataset for interfaces. As demonstrated in the present work, such GAP-based datasets can be directly fed into ACE training, followed by further domain-specific iterations. In addition, uncertainty-based sampling for active learning can be used to obtain a more comprehensive training dataset⁸⁰.”

Reviewer #3

The manuscript presents a significant advancement in the atomistic simulation of phase-change materials (PCM) for memory technology applications. Following a systematic approach to constructing and extending the training dataset, the authors successfully developed a computationally efficient and robust machine-learned interatomic potential based on the Atomic Cluster Expansion (ACE) framework for the Ge-Sb-Te (GST) system. This potential enabled unprecedented atomistic simulations of the GST PCM system at timescales and system sizes relevant to real devices, or, as the authors state, simulations of "the real thing." The simulation results, paired with an extensive study of the ACE model's scalability, demonstrate that the applied combination of techniques has the potential to significantly impact the field by enabling more detailed and realistic atomistic investigations of PCM device operation and performance.

While the manuscript is already of high quality and suitable for publication as it is, I have two minor points for the authors to address:

1. The ACE model used in this work has the capability to estimate the uncertainty of its predictions. Specifically, for a given atomic configuration, the model can indicate whether its prediction falls within an extrapolation or interpolation regime. Did the authors use this feature during simulations or during the preparation of the training dataset? If not, what was the rationale?

Response: We thank the reviewer for raising this insightful question. Indeed, ACE potentials offer built-in uncertainty indicators, enabling the classification of atomic configurations into interpolation and extrapolation regimes. In our work, we did not use uncertainty-based methods during the preparation of the training dataset for our ACE potential; instead, we relied on a multistep iterative training approach, which combines standard and domain-specific iterations to capture the diverse structural motifs relevant to GST device operations^{R21}. This protocol has underpinned our previously reported GST-GAP-22 potential, and we show in the present work that this approach remains effective when transitioning to the ACE framework.

Nevertheless, we fully acknowledge the usefulness of the uncertainty assessment provided by ACE potentials. In response to the reviewer's comments, we computed the uncertainty indicator for atomic configurations sampled during the 20-ns device-scale SET simulation (Figs. R1a–b). Using a threshold value ($\gamma = 1^{\text{R22}}$) to classify atomic environments into interpolation or extrapolation regimes, we found that $\approx 99.998\%$ of the atomic environments fell within the interpolation regime of our GST-ACE-24 potential.

Figure R2 (included as new Supplementary Fig. 4). The structural evolution during the 20-ns SET process simulation, quantified using (a) per-atom crystallinity, \bar{k} , and (b) per-atom uncertainty of the ACE prediction, indicated by the extrapolation grade γ (see Ref. R22). Higher γ values indicate greater model uncertainty about the corresponding local atomic motifs. (c) Histogram of the per-atom γ values of all atoms during the 20-ns SET process, drawn in a style similar to Ref. R9. A threshold value of $\gamma = 1$ (dashed line) separates interpolative ($\gamma \leq 1$) from extrapolative ($\gamma > 1$) regimes^{R22}. An example of a highly uncertain local motif is shown on the right, where all atomic γ values are greater than 1.

While the authors employed a justified approach for selecting relevant structures for the training dataset, uncertainty indication provided by the ACE model could further facilitate this process. For example, it could identify configurations that are furthest from previously observed data, thereby improving the efficiency of structural space exploration. And overall, uncertainty indication provides an additional mechanism to verify the model's transferability.

Response: We agree that uncertainty-based indicators could further improve the efficiency and robustness of exploration of configurational space. In fact, our analysis also revealed a very small fraction of atomic environments ($\approx 0.002\%$) that exceeded the interpolation capability of the current ACE potential (Fig. R2c). We note that uncertainty-based methods are of great use when searching for more complex scenarios, such as phase transitions under external electric fields or interfaces between GST and surrounding materials in devices, where unusual atomic motifs may emerge. In fact, the uncertainty indicator has been combined with active-learning strategy to improve the accuracy and robustness of an ACE potential for chemically and structurally complex systems, such as water^{R8}, metal–organic frameworks^{R9}, and binary Si–O phases^{R11}. Therefore, we

believe that uncertainty-based active learning approaches hold promise for future ACE potential development, complementing our domain-specific iterative protocol.

Action taken: We added the following text in the revised manuscript (p. 12)

“We analysed this 20-ns SET simulation using the ACE extrapolation grade⁸⁰, γ , which allows one to classify atomic configurations into interpolation and extrapolation regimes (Supplementary Fig. 4). Almost all atomic environments fell comfortably within the interpolation regime of our GST-ACE-24 potential, indicating that the nucleation and growth processes are accurately captured by our ACE model.”

Furthermore, we included the analysis of the 20-ns SET simulation using the ACE extrapolation grade as new Supplementary Figure 4 (p. S10). We also now mention the expected usefulness of uncertainty-based active learning for developing future ML potentials for GST devices with complex interfaces (p. 20).

2. *After simulating the re-crystallization of GST (the set process), the authors report an average grain size of approximately 4.6 nm. Is there a reference available (e.g., experimental measurement) to which this result can be compared? Comparing structural parameters could help establish a closer connection between simulation conditions and real material behavior.*

Response: We thank the reviewer for this important point. Indeed, Park et al. performed in situ transmission electron microscopy (TEM) experiments and showed that amorphous GST thin films crystallised into 5-nm-sized grains of the rock-salt-like cubic phase upon heating to around 150 °C^{R23}. Our simulated grain size is therefore similar to the experimentally measured grain size upon crystallisation. But we note that further thermal annealing results in grain coalescence and development, and the diameter of cubic-phase grain could reach tens of nanometres. In devices, amorphous GST also crystallises to form the same metastable cubic structure using SET pulses of tens of ns, but no grain-size data were clearly provided via the cross-sectional TEM experiments on common GST devices. Therefore, we prefer to keep the discussion on grain size concise here.

Action taken: We added the following text, together with a new reference for the experimentally reported grain sizes in recrystallised GST thin films, in the revised manuscript (p. 13):

“..., consistent with the experimentally measured grain size in GST thin films using *in situ* transmission electron microscopy⁸¹.”

Reviewer #4

Response: We thank the reviewer for co-reviewing this work.

In summary, we thank all reviewers again for their detailed and valuable feedback, which has allowed us to improve the manuscript further.

References

- R1. Dong, W. *et al.* Tunable mid-infrared phase-change metasurface. *Adv. Opt. Mater.* **6**, 1701346 (2018).
- R2. Wang, D. *et al.* Non-volatile tunable optics by design: From chalcogenide phase-change materials to device structures. *Mater. Today* **68**, 334–355 (2023).
- R3. Hosseini, P., Wright, C. D. & Bhaskaran, H. An optoelectronic framework enabled by low-dimensional phase-change films. *Nature* **511**, 206–211 (2014).
- R4. Du, K.-K. *et al.* Control over emissivity of zero-static-power thermal emitters based on phase-changing material GST. *Light Sci. Appl.* **6**, e16194 (2017).
- R5. Ríos, C. *et al.* Integrated all-photonic non-volatile multi-level memory. *Nat. Photon.* **9**, 725–732 (2015).
- R6. Ibrahim, E., Lysogorskiy, Y., Mrovec, M. & Drautz, R. Atomic cluster expansion for a general-purpose interatomic potential of magnesium. *Phys. Rev. Mater.* **7**, 113801 (2023).
- R7. Bienvenu, B. *et al.* Development of an atomic cluster expansion potential for iron and its oxides. *npj Comput. Mater.* **11**, 81 (2025).
- R8. Ibrahim, E., Lysogorskiy, Y. & Drautz, R. Efficient parametrization of transferable atomic cluster expansion for water. *J. Chem. Theory Comput.* **20**, 11049–11057 (2024).
- R9. Nicholas, T. C. *et al.* The structure and topology of an amorphous metal-organic framework. Preprint at arXiv:2503.24367 (2025).
- R10. Qamar, M., Mrovec, M., Lysogorskiy, Y., Bochkarev, A. & Drautz, R. Atomic cluster expansion for quantum-accurate large-scale simulations of carbon. *J. Chem. Theory Comput.* **19**, 5151–5167 (2023).
- R11. Erhard, L. C., Rohrer, J., Albe, K. & Deringer, V. L. Modelling atomic and nanoscale structure in the silicon–oxygen system through active machine learning. *Nat. Commun.* **15**, 1927 (2024).
- R12. Zhou, Y., Elliott, S. R., Toit, D. F. T. du, Zhang, W. & Deringer, V. L. The pathway to chirality in elemental tellurium. Preprint at arXiv:2409.03860 (2024).
- R13. Thomas du Toit, D. F., Zhou, Y. & Deringer, V. L. Hyperparameter optimization for atomic cluster expansion potentials. *J. Chem. Theory Comput.* **20**, 10103–10113 (2024).
- R14. Zuo, Y. *et al.* Performance and Cost Assessment of Machine Learning Interatomic Potentials.

- J. Phys. Chem. A* **124**, 731–745 (2020).
- R15. Liu, Y. *et al.* An automated framework for exploring and learning potential-energy surfaces. Preprint at arXiv:2412.16736 (2024).
- R16. Cheng, H. Y. *et al.* Atomic-level engineering of phase change material for novel fast-switching and high-endurance PCM for storage class memory application. In *2013 IEEE International Electron Devices Meeting* 30.6.1-30.6.4 (2013). DOI: 10.1109/IEDM.2013.6724726.
- R17. Cheng, H.-Y., Carta, F., Chien, W.-C., Lung, H.-L. & BrightSky, M. J. 3D cross-point phase-change memory for storage-class memory. *J. Phys. D: Appl. Phys.* **52**, 473002 (2019).
- R18. Rao, F. *et al.* Reducing the stochasticity of crystal nucleation to enable subnanosecond memory writing. *Science* **358**, 1423–1427 (2017).
- R19. Waldecker, L. *et al.* Time-domain separation of optical properties from structural transitions in resonantly bonded materials. *Nat. Mater.* **14**, 991–995 (2015).
- R20. Feldmann, J. *et al.* Calculating with light using a chip-scale all-optical abacus. *Nat. Commun.* **8**, 1256 (2017).
- R21. Zhou, Y., Zhang, W., Ma, E. & Deringer, V. L. Device-scale atomistic modelling of phase-change memory materials. *Nat. Electron.* **6**, 746–754 (2023).
- R22. Lysogorskiy, Y., Bochkarev, A., Mrovec, M. & Drautz, R. Active learning strategies for atomic cluster expansion models. *Phys. Rev. Mater.* **7**, 043801 (2023).
- R23. Park, Y. J., Lee, J. Y. & Kim, Y. T. In situ transmission electron microscopy study of the nucleation and grain growth of Ge₂Sb₂Te₅ thin films. *Appl. Surf. Sci.* **252**, 8102–8106 (2006).